

# Anomalies for Galilean fields

**Kristan Jensen**

C.N. Yang Institute for Theoretical Physics,
SUNY Stony Brook, Stony Brook, NY 11794-3840

## Abstract

We initiate a systematic study of 't Hooft anomalies in Galilean field theories, focusing on two questions therein. In the first, we consider the non-relativistic theories obtained from a discrete light-cone quantization (DLCQ) of a relativistic theory with flavor or gravitational anomalies. We find that these anomalies survive the DLCQ, becoming mixed flavor/boost or gravitational/boost anomalies. We also classify the pure Weyl anomalies of Schrödinger theories, which are Galilean conformal field theories (CFTs) with $z = 2$. There are no pure Weyl anomalies in even spacetime dimension, and the lowest-derivative anomalies in odd dimension are in one-to-one correspondence with those of a relativistic CFT in one dimension higher. These results classify many of the anomalies that arise in the field theories dual to string theory on Schrödinger spacetimes.



# 1 Introduction

Anomalous global symmetries provide one of the most useful handles on non-perturbative field theory. Their utility stems largely from being simultaneously exact, calculable, and a universal feature across the space of field theories. Anomalies must be matched across scales, and so give stringent checks on renormalization group flows and dualities. Furthermore, the language of anomalies and anomaly inflow is the natural one to discuss and classify topologically non-trivial phases of matter.

For all of these reasons, we would like to better understand anomalies in non-relativistic (NR) field theories, for which little is presently known.[1] Indeed, the role of anomalies in topologically non-trivial phases is almost entirely discussed in terms of the anomalies of relativistic field theory. This seems at best dubious to us.

In this note we begin a proper classification of anomalies in Galilean field theories.[2] We focus entirely on Galilean theories, as they possess more symmetry than a generic NR system and moreover the potentially anomalous symmetries are completely understood. The end result [5] after much history [6–11] is that Galilean theories couple to a version of Newton-Cartan (NC) geometry, and the symmetries one demands are invariance under coordinate reparameterizations, gauge transformations for the particle number and any other global symmetries, and a shift known in the NC literature (see e.g. [6]) as a Milne boost. The latter is the difference between Galilean theories and NR theories without a boost symmetry.[3] This suite of background geometry and symmetries satisfies a number of checks as summarized in [5], and moreover can be obtained by carefully taking the NR limit of relativistic theories [17].

Rather than performing a complete analysis, we elect to answer two basic questions. First, one natural route to an anomalous Galilean theory is to start with a relativistic theory with flavor and/or gravitational anomalies and perform a discrete light-cone quantization (DLCQ), i.e. to put the relativistic theory on a background with a lightlike circle. The dimensionally reduced theory is Galilean-invariant. Does it also have an anomaly? We find that the answer is "yes," insofar as there is no local counterterm which can render the NR theory invariant under all symmetries. Moreover, we show that the anomalies descend to mixed flavor/Milne or gravitational/Milne anomalies. They are "mixed" in the sense that one can arrange for the NR theory to be invariant under one symmetry or the other, but not both simultaneously.

Second, we consider Schrödinger theories, that is Galilean CFTs whose global symmetries in flat space comprise the Schrödinger group. We classify the pure Weyl anomalies of these theories, in analogy with the Weyl or trace anomaly of relativistic CFT, using consistency properties of field theory. The easiest way to perform this analysis is to lift the NC data to an ordinary Lorentzian metric in one more dimension with a null circle. We find that the NR Weyl anomaly takes the same form as the ordinary relativistic Weyl anomaly built from this higher-dimensional metric. For example, $2+1$-dimensional Schrödinger theories have two central charges, one which is formally analogous to $a$ in four-dimensional CFT, and the other to $c$.

Our analysis has one caveat: we only classify anomalous variations with at most $d+1$ derivatives. We expect that there are Weyl anomalies with more derivatives, and we conjecture below that they are all Weyl-covariant.

A corollary of this result is the following. The DLCQ of a relativistic CFT is a Schrödinger theory, and the Weyl anomaly of the relativistic parent survives the DLCQ to become the NR

---

[1] For example, consider a Hall system in two spatial dimensions for which a gravitational Chern-Simons term for a $SO(2)$ spin connection appears in the low-energy effective action. Despite much ink spilled, it is not known if the boundary field theory has a corresponding anomaly, or if a boundary counterterm cancels the variation of the Chern-Simons term.

[2] There has been some work classifying pure Weyl [1–3] and axial anomalies [4] in Lifshitz theories.

[3] See [12–16] for other perspectives on the local symmetries of Galilean theories.

Weyl anomaly. This in turn gives a prediction for the Weyl anomaly of Schrödinger theories holographically dual to string theory on so-called Schrödinger spacetimes [18,19] with $z = 2$. It would be nice to see this directly in holography using the dictionary (see [20, 21]) which allows for the field theory to couple to a curved geometry.

Similarly, our results for flavor and gravitational anomalies hold for the string theory embeddings of Schrödinger holography [22–24], wherein the NR field theory is obtained by DLCQ together with a holonomy for a global symmetry around the null circle.

The rest of this note is organized as follows. In the next Section, we summarize the details of Newton-Cartan geometry we require, and its relation to null reductions. We go on in Section 3 to show that flavor and gravitational anomalies survive DLCQ. Finally, we classify pure Weyl anomalies in Schrödinger theories in Section 4.

*Note:* Since this note appeared on the arXiv there have been a number of other papers regarding anomalies in Galilean field theories. We would like to highlight two developments. First, the authors of [25,26] independently classified the potential Weyl anomalies of Galilean theories and their findings reproduced our findings in Section 4, in particular that $z = 2$ Schrödinger theories in odd spacetime dimension have a single *A*-type Weyl anomaly. Second, there have been several attempts [27–29] to compute the Weyl anomaly of a non-relativistic free field in $2 + 1$-dimensions. This anomaly should satisfy a sort of sum rule. Compactifying a free $3 + 1$-dimensional relativistic scalar on a spacetime with a lightlike circle of coordinate periodicity $2\pi$, one obtains a tower of decoupled non-relativistic scalars of masses $m = n$ for all $n \in \mathbb{Z}$. Summing up the anomalies of this tower, one must recover the anomaly of the relativistic parent. Of the various computations in the literature, only that of [28] passes this test. Those authors find that only a massless non-relativistic scalar has an anomaly, and go on to demonstrate that this result remains true to all orders in perturbation theory.

## 2 Preliminaries

We presently review the machinery which we require for the rest of this note.

### 2.1 Newton-Cartan geometry

The version of Newton-Cartan geometry which will be useful for us is the following. A Newton-Cartan structure (see e.g. [7,30]) on a $d$-dimensional spacetime $\mathcal{M}_d$ is comprised of a one-form $n_\mu$, a symmetric, positive-semi-definite rank $d - 1$ tensor $h_{\mu\nu}$, and a $U(1)$ connection $A_\mu$. The tensors $(n_\mu, h_{\mu\nu})$ are almost arbitrary: we require that

$$\gamma_{\mu\nu} = n_\mu n_\nu + h_{\mu\nu}, \tag{1}$$

is positive-definite. These tensors algebraically determine the upper-index data $v^\mu$ and $h^{\mu\nu}$ satisfying

$$v^\mu n_\mu = 1, \qquad h_{\mu\nu} v^\nu = 0, \qquad h^{\mu\nu} n_\nu = 0, \qquad h^{\mu\rho} h_{\nu\rho} = \delta^\mu_\nu - v^\mu n_\nu. \tag{2}$$

The "velocity vector" $v^\mu$ defines a local time direction, and $h_{\mu\nu}$ gives a metric on spatial slices. (NC geometry with general $n$ not closed was only studied recently [5,31].)

We can define a covariant derivative using the tensors that make up the NC structure. Unlike in Riemannian geometry where there is essentially one derivative that can be defined with a metric, there are many possible derivatives that can be defined from the NC data. One choice of connection is [5]

$$\Gamma^\mu{}_{\nu\rho} = v^\mu \partial_\rho n_\nu + \frac{1}{2} h^{\mu\sigma} \left( \partial_\nu h_{\rho\sigma} + \partial_\sigma h_{\nu\rho} - \partial_\sigma h_{\nu\rho} \right) + h^{\mu\sigma} n_{(\nu} F_{\rho)\sigma}, \tag{3}$$

where the brackets indicate symmetrization with weight $1/2$ and $F$ is the field strength of $A_\mu$. The corresponding derivative $D_\mu$ has the nice feature that

$$D_\mu n_\nu = 0, \qquad D_\mu h^{\nu\rho} = 0. \tag{4}$$

Galilean field theories naturally couple to this sort of NC geometry [5, 10, 11], where $A_\mu$ is the gauge field which couples to particle number. For instance, the action of a free Galilean scalar $\varphi$ carrying charge $m$ under particle number is

$$S_{free} = \int d^d x \sqrt{\gamma} \left\{ \frac{i\nu^\mu}{2} \left( \varphi^\dagger D_\mu \varphi - (D_\mu \varphi^\dagger)\varphi \right) - \frac{h^{\mu\nu}}{2m} \varphi^\dagger \varphi \right\}, \tag{5}$$

with $D_\mu \varphi = (\partial_\mu - imA_\mu)\varphi$. Note that we have used $\gamma_{\mu\nu}$ defined in (1) to define an invariant measure $d^d x \sqrt{\gamma}$. In coupling a Galilean theory to NC geometry, one demands invariance under (i.) reparameterization of coordinates, (ii.) $U(1)$ gauge transformations, and (iii.) shift transformations known in the NC literature [6] as Milne boosts. Under the boost, the tensors $(n_\mu, h^{\mu\nu})$ are invariant and $(\nu^\mu, h_{\mu\nu}, A_\mu)$ shift as

$$\nu^\mu \to \nu^\mu + \psi^\mu, \quad h_{\mu\nu} \to h_{\mu\nu} - (n_\mu \psi_\nu + n_\nu \psi_\mu) + n_\mu n_\nu \psi^2, \quad A_\mu \to A_\mu + \psi_\mu - \frac{1}{2} n_\mu \psi^2, \tag{6}$$

Here, $\psi_\mu$ is spatial, meaning $\nu^\mu \psi_\mu = 0$, and we have used the shorthand $\psi^\mu = h^{\mu\nu}\psi_\nu$, $\psi^2 = \psi^\mu \psi_\mu$. One can easily verify that (5) is invariant under the Milne boosts.

The boosts are crucial: they impose a covariant version of the Galilean boost invariance. However, it is troublesome to obtain tensors which are invariant under both the boosts and $U(1)$ gauge invariance. For example, the connection we defined above (3) is gauge-invariant, but not Milne-invariant [5]. One can define another connection which is Milne-invariant, but the resulting derivative is not gauge-invariant. There is no connection which is invariant under both symmetries.[4] As a result, the covariant derivative of an boost and gauge-invariant tensor is not a boost and gauge-invariant tensor. C'est la vie, but this fact complicates the classification of potential anomalies.

In the absence of any anomalies, $W$ is invariant under infinitesimal coordinate transformations, gauge transformations, and Milne boosts. These symmetries lead to (potentially anomalous) Ward identities [5, 11]. From the generating functional $W$ of correlation functions, one defines a sort of stress tensor complex. Letting $W$ depend on an overcomplete parameterization of the background, $W = W[n_\mu, \nu^\mu, h^{\mu\nu}, A_\mu]$, we define the number current $J^\mu$, momentum current $\mathscr{P}_\mu$, energy current $\mathscr{E}^\mu$, and spatial stress tensor $T_{\mu\nu}$ via

$$\delta W = \int d^d x \sqrt{\gamma} \left\{ \delta A_\mu J^\mu - \delta \bar{\nu}^\mu \mathscr{P}_\mu - \delta n_\mu \mathscr{E}^\mu - \frac{\delta \bar{h}^{\mu\nu}}{2} T_{\mu\nu} \right\}. \tag{7}$$

Here we let the variations of $n_\mu$ be completely arbitrary, in which case some of the variations of $\nu^\mu$ and $h^{\mu\nu}$ are fixed in terms of $\delta n_\mu$, e.g.

$$\delta \nu^\mu = -\nu^\mu \nu^\nu \delta n_\nu + P^\mu_\nu \delta \bar{\nu}^\nu, \tag{8}$$

where $\delta \bar{\nu}^\mu$ is arbitrary, and similarly for $\delta \bar{h}^{\mu\nu}$. The $U(1)$ gauge invariance implies that $J^\mu$ is conserved, the Milne invariance equates momentum with the spatial part of the particle number current, $\mathscr{P}_\mu = h_{\mu\nu} J^\nu$, and the coordinate reparameterization invariance leads to conservation equations for the energy current and spatial stress tensor.

---

[4]If one has a gauge and Milne-invariant vector $\mathfrak{v}^\mu$ satisfying $\mathfrak{v}^\mu n_\mu > 0$ everywhere, then one can use this vector to build a boost and gauge-invariant derivative [32].

When the theory has "local" anomalies, $W$ has an infinitesimal variation under at least one of these transformations, and the anomaly may be characterized by the variation $\delta_\chi W \neq 0$. Equivalently, the anomaly may be characterized by the modified, anomalous Ward identities.

A Galilean CFT is also invariant (up to anomalies) under "Weyl" rescalings which are characterized by a critical exponent $z$, under which the background fields rescale as

$$n_\mu \to e^{z\Omega} n_\mu, \qquad h_{\mu\nu} \to e^{2\Omega} h_{\mu\nu}, \qquad A_\mu \to e^{(2-z)\Omega} A_\mu. \tag{9}$$

The case $z = 2$ is special, and in that case the CFT is called a "Schrödinger" theory, as its global symmetries in flat space form the Schrödinger group. In Section 4 we will consider Weyl anomalies for Schrödinger theories. In that case, denoting the Weyl variation of $W$ as

$$\delta_\Omega W = \int d^d x \sqrt{\gamma} \, \delta\Omega \, \mathscr{A}, \tag{10}$$

the anomalous Weyl Ward identity is

$$2n_\mu \mathscr{E}^\mu - h^{\mu\nu} T_{\mu\nu} = \mathscr{A}. \tag{11}$$

## 2.2 Null reduction

This geometric structure could have been (and perhaps should have been) anticipated from DLCQ. Recall that one way to construct a $d$-dimensional Galilean-invariant theory is to start with a relativistic theory in $d + 1$ dimensional Minkowski space in light-cone coordinates,

$$g = 2dx^0 dx^- + d\vec{x}^2,$$

and compactify the null coordinate $x^-$ with some radius $R$. On purely algebraic grounds – the Poincaré generators which commute with $P_-$ generate the Galilean algebra, with $P_-$ playing the role of particle number – the $d$-dimensional theory one obtains on $(x^0, \vec{x})$ is Galilean-invariant. Similarly, if one starts with a relativistic CFT, the lower-dimensional theory is invariant under the Schrödinger group [22], which is generated by the Galilean algebra in addition to a dilatation and special conformal generator.

It is well known that DLCQ is subtle (see e.g. [22, 23] and references therein). The zero-modes of the null reduction have to be treated carefully, and this is not always understood. However we are only interested in the symmetries of the problem, rather than a careful definition of the ensuing Galilean theory, and so these subtleties are irrelevant for us.

It is worth noting that the string theoretic realizations of Schrödinger field theories in holography either arise from DLCQ, or from DLCQ in addition to a holonomy for a global symmetry around the null circle [22–24].

Of course, we could consider the most general DLCQ. That is, we could put a relativistic theory on the most general $d + 1$-dimensional spacetime with a null isometry,

$$g = 2n_\mu dx^\mu (dx^- + A) + h_{\mu\nu} dx^\mu dx^\nu, \tag{12}$$

where the component functions depend on the $x^\mu$ but not $x^-$ and $h_{\mu\nu}$ is a rank $d - 1$ positive-semi-definite tensor. The null isometry is

$$n^M \partial_M = \partial_-. \tag{13}$$

After reducing on $x^-$, the lower-dimensional theory is a Galilean theory which couples to $(n_\mu, h_{\mu\nu}, A_\mu)$, which we recognize as the defining data of a NC structure. Moreover, all of the symmetries we outlined in the previous Subsection are manifest here. The gauge field $A_\mu$ is just the graviphoton of the reduction, and its $U(1)$ gauge invariance just corresponds to additive

reparameterizations of $x^-$. The Milne boosts correspond to an ambiguity in the extraction of $(h_{\mu\nu}, A_\mu)$ from $g$, as discussed in [5]. One can redefine $A$ and $h$ as

$$A_\mu \to A_\mu + \Psi_\mu, \qquad h_{\mu\nu} \to h_{\mu\nu} - (n_\mu \Psi_\nu + n_\nu \Psi_\mu), \tag{14}$$

for any one-form $\Psi_\mu(x^\nu)$, which leaves $g$ unchanged. In order for $h_{\mu\nu}$ to remain rank $d-1$, however, we must have

$$\Psi_\mu = \psi_\mu - \frac{1}{2} n_\mu \psi^2, \qquad v^\mu \psi_\mu = 0. \tag{15}$$

But these are just the Milne boosts (6). And of course $d$-dimensional reparameterizations are just $d+1$-dimensional reparameterizations along fibers of constant $x^-$.

So DLCQ automatically gives NR theories coupled to NC geometry, invariant under the symmetries above.

Later, it will be important that there is no one-form $dx^- + V_\mu(x^\nu)dx^\mu$ which is invariant under both the $U(1)$ particle number symmetry and the Milne boosts: $dx^-$ is boost-invariant, but not $U(1)$-invariant, while $dx^- + A$ is $U(1)$-invariant but not boost-invariant.

Historically, some of these results were known some time ago, although from a very different perspective. It was first recognized in [7] (see also [33]) that a NC structure can be obtained via null reduction of a Lorentzian $d+1$ dimensional manifold, although these authors restricted $n$ to be closed. Of course this reduction still works if $dn \neq 0$ [5, 31]. The role of the Milne boosts in $d+1$ dimensions was only realized in [5].

Aside from its obvious importance in DLCQ, the null reduction (12) will be very useful in what follows, even for NR theories which do not follow from DLCQ. Its primary virtue for us is that all of the NC symmetries are manifest therein. So we can efficiently construct tensors under the NC symmetries by constructing tensors on an auxiliary $d+1$-dimensional spacetime with metric (12). For instance, using the Levi-Civita connection

$$(\Gamma_g)^M{}_{NP} = \frac{1}{2} g^{MQ} \left( \partial_N g_{PQ} + \partial_P g_{NQ} - \partial_Q g_{NP} \right), \tag{16}$$

obtained from $g_{MN}$ to define the $d+1$-dimensional covariant derivative $\mathscr{D}_M$, we can define the Riemann tensor

$$\mathscr{R}^M{}_{NPQ} = \partial_P (\Gamma_g)^M{}_{NQ} - \partial_Q (\Gamma_g)^M{}_{NP} + (\Gamma_g)^M{}_{SP} (\Gamma_g)^S{}_{NQ} - (\Gamma_g)^M{}_{SQ} (\Gamma_g)^S{}_{NP}, \tag{17}$$

and so $\mathscr{R} = g^{MN} \mathscr{R}^P{}_{MPN}$. Because $\mathscr{R}$ is invariant under all of the higher-dimensional symmetries, it gives an invariant $d$-dimensional scalar under all of the NC symmetries.

As we mentioned above, the flat-space DLCQ of a relativistic CFT gives a Schrödinger theory. So it should not be a surprise that the Weyl symmetry of a Schrödinger theory nicely fits into the null reduction (12). Note that the Weyl rescaling of $g$

$$g \to e^{2\Omega} g \tag{18}$$

is equivalent to the NR $z = 2$ Weyl rescaling of $n_\mu$ and $h_{\mu\nu}$ in (9). This will be useful for us when we consider pure Weyl anomalies in Schrödinger theories in Section 4.

As a simple example, consider the DLCQ of a conformally coupled complex scalar $\Phi$

$$S_R = -\frac{1}{4\pi} \int d^{d+1}x \sqrt{-g} \left\{ g^{MN} \partial_M \Phi^\dagger \partial_N \Phi + \xi \mathscr{R} \Phi^\dagger \Phi \right\}, \qquad \xi = \frac{d-1}{4d}, \tag{19}$$

and compactify $x^-$ with periodicity $2\pi$. This theory is Weyl-invariant provided that $\Phi$ transforms with weight $\frac{1-d}{2}$. Expanding $\Phi$ in Fourier modes of $x^-$, $\Phi = \sum_n \varphi_n(x^\mu) e^{inx^-}$, the relativistic action $S_R$ becomes (using that $\sqrt{-g} = \sqrt{\gamma}$)

$$S_R = \sum_n \int d^d x \sqrt{\gamma} \left\{ \frac{inv^\mu}{2} \left( \varphi_n^\dagger D_\mu \varphi_n - (D_\mu \varphi_n^\dagger) \varphi_n \right) - \frac{h^{\mu\nu}}{2} D_\mu \varphi_n^\dagger D_\nu \varphi_n - \frac{\xi \mathscr{R}}{2} \varphi_n^\dagger \varphi_n \right\}, \tag{20}$$

with $D_\mu \varphi_n = (\partial_\mu - inA_\mu)\varphi_n$. The term for each $n \neq 0$ is just $n$ times the action of the NR analogue of a conformally coupled scalar $\varphi_n$ carrying charge $n$ under particle number [5,34]. The zero-mode action is also invariant under the local Galilean and Weyl symmetries, but it is obvious that it must be treated carefully when computing observables.

# 3 Flavor and/or gravitational anomalies from DLCQ

Suppose we study the NR theory that arises from the DLCQ of a relativistic theory with flavor and/or gravitational anomalies.[5] Does the NR theory have anomalies?

We proceed in two steps. First, we efficiently represent the anomalies of the underlying relativistic theory with anomaly inflow. Then we put the relativistic theory on a background with a null isometry (12), whereby we express the anomalies in terms of the NC data to which the NR theory couples. This variation can be cancelled off by adding a counterterm built from the NC data. However said counterterm is not boost-invariant.

It is easiest to work with this boost-non-invariant description. We find that this boost variation cannot be removed by a further counterterm, and so is a genuine anomaly which we then write in terms of an anomaly inflow. So the anomalies of the relativistic parent descend to mixed flavor/boost or gravitational/boost anomalies in the NR theory, and the counterterm we construct is a "Bardeen counterterm" which shifts the anomaly from the flavor or gravitational sector to being a boost anomaly.

## 3.1 Anomaly inflow

Perhaps the simplest way to describe flavor and gravitational anomalies in relativistic QFT is via the anomaly inflow mechanism [35]. Given a field theory on a $D$-dimensional spacetime $\mathcal{M}_D$ coupled to a flavor gauge field $\mathcal{A}_M$ and Riemannian metric $g_{MN}$, the local anomalies are encoded in a $D+2$-form $\mathcal{P}$ known as the anomaly polynomial. $\mathcal{P}$ is built from the Chern classes of the field strength $\mathcal{F}_{MN}$ of $\mathcal{A}_M$ and Pontryagin classes of the Riemann curvature $\mathcal{R}^M{}_{NPQ}$, and so $d\mathcal{P} = 0$.

$\mathcal{P}$ determines the variation of the field theory generating functional $W$ through the descent equations. $\mathcal{P}$ is the exterior derivative of a Chern-Simons $D+1$-form $I$, $\mathcal{P} = dI$, whose gauge and/or coordinate variation is a derivative of the $d$-form $G_\chi$

$$\delta_\chi I = d G_\chi \,. \tag{21}$$

This determines the anomalous variation of $W$ via

$$\delta_\chi W = -\int_{\mathcal{M}_D} G_\chi \,, \tag{22}$$

or equivalently, one constructs

$$W_{cov} = W + \int_{\mathcal{M}_{D+1}} I \,, \tag{23}$$

which is invariant under all symmetries, where we have extended the gauge field and metric on $\mathcal{M}_D$ to a gauge field and metric on the $D+1$-dimensional manifold $\mathcal{M}_{D+1}$ which has $\mathcal{M}_D$ as its boundary. (23) neatly encodes the idea of anomaly inflow: we imagine that our field theory lives on the boundary of a "Hall" system whose action is the Chern-Simons term $\int I$, and the

---

[5]This includes those theories which are dual to string theory on asymptotically Schrödinger backgrounds, where there are bulk Chern-Simons terms.

$$\mathcal{M}_d \xleftarrow{\text{reduce on } x^-} \mathcal{M}_{d+1} = \partial \mathcal{M}_{d+2} \xleftarrow{\text{inflow}} \mathcal{M}_{d+2} \xleftarrow{\text{descent: } I \leftarrow \mathcal{P}} \mathcal{M}_{d+3}$$

Figure 1: The short sequence which relates the anomalies of a relativistic theory to those of the $d$-dimensional NR theory realized by DLCQ. The relativistic parent lives on $\mathcal{M}_{d+1}$, and its null reduction leads to the NR theory on $\mathcal{M}_d$. The anomalies of the parent are described via inflow by letting $\mathcal{M}_{d+1}$ be the boundary of a $d+2$-manifold $\mathcal{M}_{d+2}$ which we equip with a Chern-Simons form $I$. The anomaly polynomial $\mathcal{P}$ is a formal $d+3$ form which may be thought of as living on a formal $\mathcal{M}_{d+3}$.

anomalies of the boundary theory arise because currents or energy-momentum can flow from the "bulk theory" on $\mathcal{M}_{D+1}$ into the boundary.

The anomaly inflow mechanism is also at play in holography. When an anomalous field theory has a gravity dual, the metric and gauge field in the field theory are the asymptotic values of a dynamical bulk metric and gauge field, and the gravitational theory has a Chern-Simons term $I$ and the field theory anomaly polynomial is $\mathcal{P} = dI$.

## 3.2 Constructing the counterterm

Now consider the $d$-dimensional Galilean theory on $\mathcal{M}_d$ obtained by performing the most general DLCQ of a relativistic theory on $\mathcal{M}_{d+1}$ with anomaly polynomial $\mathcal{P}$. That is, the relativistic theory is coupled to a $d+1$-dimensional metric $g_{MN}$ and (not necessarily abelian) flavor gauge field $\mathscr{A}_M$ with a null symmetry along $x^-$,

$$g = 2n_\mu dx^\mu (dx^- + A) + h_{\mu\nu} dx^\mu dx^\nu, \qquad \mathscr{A} = \hat{\mathscr{A}}_\mu dx^\mu + \mathscr{A}_-(dx^- + A), \qquad (24)$$

where the component fields depend on $x^\mu$ but not $x^-$, and implicitly $x^-$ is compactified with some radius. The $(n_\mu, h_{\mu\nu}, A_\mu)$ become the NC structure to which the $d$-dimensional NR theory couples, $\hat{\mathscr{A}}_\mu$ becomes a background flavor gauge field which couples to the flavor symmetry of the NR theory, and $\mathscr{A}_-$ becomes a scalar source. The anomalies of the $d+1$-dimensional theory are most efficiently described in terms of the Chern-Simons form $I$ on a $d+2$-dimensional spacetime $\mathcal{M}_{d+2}$ which has $\mathcal{M}_{d+1}$ as its boundary. Reducing on the null circle leads to the $d$-dimensional spacetime $\mathcal{M}_d$ on which the NR theory lives. This gives the short sequence represented in Fig. 1.

When one instead reduces a relativistic theory with anomalies on a spatial or thermal circle, the $d$-dimensional description is also relativistic. In that case one can construct a local counterterm built out of the $d$-dimensional background which cancels the anomalous variation. That such a counterterm exists is a corollary of the fact that there are no flavor or gravitational anomalies in odd-dimensional relativistic theories. It turns out that we can construct such a counterterm in the null case, although we lack an a priori argument that this should be so. However, said counterterm is not invariant under Milne boosts.

If one is interested just in the lower-dimensional theory obtained from a spatial circle reduction, the precise form of this counterterm is uninteresting. However, for thermal circles, this counterterm is physical: it is intimately related to non-renormalized anomaly-induced transport. See [36] (as well as [37]) for details. For an arbitrary anomaly polynomial, the requisite counterterm was constructed in [38] (see also [39]) using the technology of transgression forms, which will also be useful here.

The basic idea can be illustrated with a pure flavor anomaly. From the point of view of the NR theory, $\mathscr{A}_-$ is a scalar source which transforms in the adjoint representation of the flavor symmetry, rather than as a component of a connection. So we can define a new flavor connection $\bar{\mathscr{A}}$ by simply subtracting off the $\mathscr{A}_-$ component. To do this in a $U(1)$-invariant

way, we subtract off a term proportional to $dx^- + A$,[6]

$$\bar{\mathscr{A}} \equiv \mathscr{A} - \mathscr{A}_-(dx^- + A) = \hat{\mathscr{A}}_\mu dx^\mu, \tag{25}$$

which we recognize as just the hatted connection from (24). This hatted connection is almost a good flavor connection under the $d$-dimensional symmetries: while $\mathscr{A}$ is Milne-invariant, $dx^- + A$ varies under the Milne boosts, and so $\hat{\mathscr{A}}$ varies as

$$\hat{\mathscr{A}} \to \hat{\mathscr{A}} - \mathscr{A}_-\Psi, \qquad \Psi_\mu = \psi_\mu - \frac{1}{2}n_\mu\psi^2, \tag{26}$$

which still has no leg along $dx^-$. In any case, it is clear that both $\hat{\mathscr{A}}$ and the field strength of $\hat{\mathscr{A}}$, $\hat{\mathscr{F}}$, have no legs along $dx^-$. (The covariant statement is that $\hat{\mathscr{F}}_{MN}n^N = 0$ and $\hat{\mathscr{A}}_M n^M$ can be set to zero by a gauge transformation.) Consequently, the Chern-Simons form $I$ evaluated on the hatted connection and field strength has no leg along $dx^-$, and so its integral on $\mathscr{M}_{d+2}$ vanishes. Denoting $I[\hat{\mathscr{A}}, \hat{\mathscr{F}}] = \hat{I}$, we trivially have

$$\int_{\mathscr{M}_{d+2}} I = \int_{\mathscr{M}_{d+2}} I - \int_{\mathscr{M}_{d+2}} \hat{I}, \tag{27}$$

which is where transgression forms become useful.

We refer the reader who is unfamiliar with the transgression machinery to the Appendix of [38] for a concise and modern discussion. Essentially, this technology is the natural way to describe the way characteristic classes, and objects like Chern-Simons forms constructed from them, depend on the connection. The result we require here is that a Chern-Simons form $I$ evaluated for two different connections $\mathscr{A}_1$ and $\mathscr{A}_2$, which we denote as $I_n = I[\mathscr{A}_n, \mathscr{F}_n]$, obeys

$$I_1 - I_2 = V_{12} + dW_{12}, \tag{28}$$

where

$$\mathscr{A}(\tau) = \mathscr{A}_2 + \tau(\mathscr{A}_1 - \mathscr{A}_2), \qquad \mathscr{F}(\tau) = d\mathscr{A}(\tau) + \mathscr{A}(\tau) \wedge \mathscr{A}(\tau),$$
$$V_{12} = \int_0^1 d\tau\,(\mathscr{A}_1 - \mathscr{A}_2) \wedge \cdot \frac{\partial \mathscr{P}(\tau)}{\partial \mathscr{F}(\tau)}, \qquad W_{12} = \int_0^1 d\tau\,(\mathscr{A}_1 - \mathscr{A}_2) \wedge \cdot \frac{\partial I(\tau)}{\partial \mathscr{F}(\tau)}. \tag{29}$$

Here $\mathscr{A}(\tau)$ interpolates between $\mathscr{A}_2$ and $\mathscr{A}_1$, $\mathscr{P}(\tau)$ and $I(\tau)$ refer to the anomaly polynomial and Chern-Simons form evaluated on it, and $\cdot$ refers to a trace over flavor indices. When $\mathscr{A}_1$ and $\mathscr{A}_2$ differ by an adjoint tensor, $V_{12}$ is a gauge-invariant form and the variations of $I_1$ and $I_2$ are carried by the boundary term $W_{12}$.

Combining this with (23) and (27), taking $\mathscr{A}_2 = \hat{\mathscr{A}}$ and $\mathscr{A}_1 = \mathscr{A}$, and denoting

$$I - \hat{I} = \hat{V} + d\hat{W}, \tag{30}$$

then we can rewrite the covariant field theory generating functional (23) as

$$W_{cov} = W + \int_{\mathscr{M}_{d+2}} I = \left(W + \int_{\mathscr{M}_{d+1}} \hat{W}\right) + \int_{\mathscr{M}_{d+2}} \hat{V} = W' + \int_{\mathscr{M}_{d+2}} \hat{V}, \tag{31}$$

where we have redefined $W$ by the local counterterm $\hat{W}$ as $W \to W + \int_{\mathscr{M}_{d+1}} \hat{W}$. Because $\mathscr{A} - \hat{\mathscr{A}} = \mathscr{A}_-(dx^- + A)$ is an adjoint tensor, $\hat{V}$ is gauge-invariant and so $W'$ is too. That is, the

---

[6]There is a fully $d + 1$-dimensionally covariant version of this construction along the lines of that presented in [38] for thermal circles. The generalization to the null case is straightforward: the covariant version of $\mathscr{A}_-$ is $n^M \mathscr{A}_M + \Lambda_K$ where $\Lambda_K$ is the flavor gauge transformation which together with $n^M$ generates the symmetry, and the covariant version of $dx^- + A$ is any one-form $u$ which obeys $n^M u_M = 1$. However any such choice breaks the Milne redundancy, and so the redefined connection varies under the Milne boosts.

local counterterm $\int \hat{W}$ renders the NR theory invariant under flavor gauge transformations. However, because $\mathscr{A} - \hat{\mathscr{A}} = \mathscr{A}_- (dx^- + A)$ varies under Milne boosts, the redefined $W'$ is now non-invariant under Milne boosts.

Before going on, we observe that (31) now looks like anomaly inflow for boosts, where $\hat{V}$ plays the role of a Chern-Simons form for Milne boosts. That is,

$$d\hat{V} = \mathscr{P} - \hat{\mathscr{P}}, \tag{32}$$

which is boost-invariant, but the boost variation of $\hat{V}$ is a total derivative,[7]

$$\delta_\psi \hat{V} = dG_\psi, \tag{33}$$

so that $W'$ varies under Milne boosts as

$$\delta_\psi W' = -\int_{\mathscr{M}_{d+1}} G_\psi. \tag{34}$$

In fact, this demonstrates that the Milne variation is a genuine anomaly. That is, there is no local $d + 1$-dimensional counterterm which can remove (34).[8] To see this, suppose that such a counterterm exists, which we denote as $\int_{\mathscr{M}_{d+1}} C$. Then $\hat{V}' \equiv \hat{V} + dC$ is invariant under all symmetries, and $d\hat{V}' = \mathscr{P} - \hat{\mathscr{P}}$. So we must be able to "transgress" the difference $\mathscr{P} - \hat{\mathscr{P}}$ in a flavor/gravitational/boost-invariant way. The most general such transgression from $\hat{\mathscr{P}}$ to $\mathscr{P}$ is to consider the most general flow $\mathscr{A}(\tau)$ which interpolates from $\hat{\mathscr{A}}$ at $\tau = 0$ and flows to $\mathscr{A}$ at $\tau = 1$. In terms of that flow one has

$$\mathscr{P} - \hat{\mathscr{P}} = d\hat{V}', \qquad \hat{V}' = \int_0^1 d\tau \, \partial_\tau \mathscr{A}(\tau) \wedge \cdot \frac{\partial \mathscr{P}(\tau)}{\partial \mathscr{F}(\tau)}. \tag{35}$$

In order for $\hat{V}'$ to be invariant, we require that each of the terms in the integral expression (35) for $\hat{V}'$ to be covariant under the symmetries. However, it is easy to see that no such $\mathscr{A}(\tau)$ exists so that $\partial_\tau \mathscr{A}(\tau)$ is covariant: the only zero-derivative covariant one-form available is $n = n_M dx^M$, and one cannot reach $\hat{\mathscr{A}}$ by adding a scalar times $n$ to $\mathscr{A}$.

So far, we have shown that the DLCQ of a theory with a pure flavor anomaly leads to a NR theory with a mixed flavor/boost anomaly. The term "mixed" refers to the fact that one can use a local counterterm to make the NR theory invariant under one symmetry or the other, but not both simultaneously. Now we consider the DLCQ of a theory with arbitrary anomalies. This is only an upgraded version of the analysis above, so we just mention the highlights.

In addition to extending the flavor connection $\mathscr{A}$ and symmetry data to higher dimensions, we also extend the metric $g$ and so the Levi-Civita connection $\Gamma_g$.[9] It is convenient to represent $\Gamma_g$ as a matrix-valued connection one-form,

$$(\Gamma_g)^M{}_N = (\Gamma_g)^M{}_{NP} dx^P, \tag{36}$$

and the Riemann curvature $\mathscr{R}^M{}_{NPQ}$ as a curvature two-form,

$$\mathscr{R}^M{}_N = \frac{1}{2} \mathscr{R}^M{}_{NPQ} dx^P \wedge dx^Q = d(\Gamma_g)^M{}_N + (\Gamma_g)^M{}_P \wedge (\Gamma_g)^P{}_N. \tag{37}$$

---

[7]We will justify both of these assertions and compute the Milne variation of $\hat{V}$ in the next Subsection.

[8]In this statement we implicitly require $d > 1$ in order to have a Milne symmetry in the first place.

[9]To ensure that the gravitational anomaly inflow is completely intrinsic, we require that the extrinsic curvature of $\mathscr{M}_{d+1} = \partial \mathscr{M}_{d+2}$ vanishes. In these coordinates that means $(\Gamma_g)^\perp{}_M = (\Gamma_g)^M{}_\perp = 0$. Alternatively, one can allow for extrinsic curvature and add a local counterterm which depends on it in such a way as to obtain the correct gravitational anomaly on $\mathscr{M}_{d+1}$.

In the coordinates in which we expressed $g_{MN}$ (24), the $-$ component of $\Gamma_g$ is a tensor

$$(\Gamma_g)^M{}_{N-} = \mathscr{D}_N n^M \,, \tag{38}$$

which we can use to define a "hatted" connection $\hat{\Gamma}_g$ along the same lines as $\hat{\mathscr{A}}$ in (25),

$$(\hat{\Gamma}_g)^M{}_N = (\Gamma_g)^M{}_N - \mathscr{D}_N n^M (dx^- + A) \,. \tag{39}$$

Both $\hat{\Gamma}_g$ and the corresponding curvature $\hat{\mathscr{R}}$ have no leg along $dx^-$. (The covariant statement is that $dx^- + A$ is a one-form $u$ obeying $n^M u_M = 1$, $\hat{\mathscr{R}}^M{}_{NPQ} n^Q = 0$, and $(\hat{\Gamma}_g)^M{}_{NP} n^P$ can be set to zero by a coordinate reparameterization.) $\hat{\Gamma}_g$ transforms as a connection under coordinate reparameterizations, but inherits a non-trivial transformation law under Milne boosts owing to the transformation of $dx^- + A$,

$$(\hat{\Gamma}_g)^M{}_N \to (\hat{\Gamma}_g)^M{}_N - \mathscr{D}_N n^M \, \Psi \,, \qquad \Psi_\mu = \psi_\mu - \frac{1}{2} n_\mu \psi^2 \,. \tag{40}$$

As above, the Chern-Simons form $I$ evaluated for the hatted connections, $\hat{I} = I[\hat{\mathscr{A}}, \hat{\mathscr{F}}, \hat{\Gamma}_g, \hat{\mathscr{R}}]$, has no leg along $dx^-$ and so its integral vanishes on $\mathscr{M}_{d+2}$, giving

$$\int_{\mathscr{M}_{d+2}} I = \int_{\mathscr{M}_{d+2}} I - \int_{\mathscr{M}_{d+2}} \hat{I} \,.$$

In analogy with (29), the difference between $I$ evaluated for two sets of connections $\{\mathscr{A}_1, \Gamma_1\}$ and $\{\mathscr{A}_2, \Gamma_2\}$ may be represented as

$$I_1 - I_2 = V_{12} + dW_{12} \,,$$

where $V_{12}$ and $W_{12}$ are transgression forms. Flowing in the space of connections as

$$\mathscr{A}(\tau) = \mathscr{A}_2 + \tau(\mathscr{A}_1 - \mathscr{A}_2) \,, \qquad \Gamma^M{}_N(\tau) = (\Gamma_2)^M{}_N + \tau\left((\Gamma_1)^M{}_N - (\Gamma_2)^M{}_N\right) \,, \tag{41}$$

we have

$$V_{12} = \int_0^1 d\tau \left\{ (\mathscr{A}_1 - \mathscr{A}_2) \wedge \cdot \frac{\partial \mathscr{P}(\tau)}{\partial \mathscr{F}(\tau)} + \left((\Gamma_1)^M{}_N - (\Gamma_2)^M{}_N\right) \wedge \frac{\partial \mathscr{P}(\tau)}{\partial \mathscr{R}^M{}_N} \right\} \,,$$
$$W_{12} = \int_0^1 d\tau \left\{ (\mathscr{A}_1 - \mathscr{A}_2) \wedge \cdot \frac{\partial I(\tau)}{\partial \mathscr{F}(\tau)} + \left((\Gamma_1)^M{}_N - (\Gamma_2)^M{}_N\right) \wedge \frac{\partial I(\tau)}{\partial \mathscr{R}^M{}_N} \right\} \,. \tag{42}$$

When $\mathscr{A}_1 - \mathscr{A}_2$ and $\Gamma_1 - \Gamma_2$ are covariant tensors, $V_{12}$ is a gauge and reparameterization-invariant form and $W_{12}$ carries the variations of $I_1$ and $I_2$. This is indeed the case when

$$\mathscr{A}_2 = \hat{\mathscr{A}} \,, \qquad \mathscr{A}_1 = \mathscr{A} \,, \qquad \Gamma_2 = \hat{\Gamma}_g \,, \qquad \Gamma_1 = \Gamma_g \,,$$

and as before we denote

$$I - \hat{I} = \hat{V} + d\hat{W} \,.$$

Putting this together with (23) we can rewrite $W_{cov}$ in the same form as (31),

$$W_{cov} = W' + \int_{\mathscr{M}_{d+2}} \hat{V} \,, \qquad W' = W + \int_{\mathscr{M}_{d+1}} \hat{W} \,.$$

As before $\hat{W}$ is a local counterterm which renders the NR theory invariant under flavor gauge transformations and coordinate reparameterizations, but leaves it anomalous under Milne boosts.

We conclude this Subsection with some simplified formulae for $\hat{V}$ and $\hat{W}$.

First, we simplify the hatted curvatures. We denote $u \equiv dx^- + A$ so $du = F$, and let $\mathscr{D}$ be the exterior covariant derivative. We also decompose the ordinary curvatures into components which are longitudinal and transverse to $n^M$,

$$\mathscr{F} = \mathscr{E} \wedge u + \mathscr{B}, \qquad \mathscr{R}^M{}_N = (\mathscr{E}_R)^M{}_N \wedge u + (\mathscr{B}_R)^M{}_N, \tag{43}$$

where

$$\mathscr{E}_M = \mathscr{F}_{MN} n^N, \quad \mathscr{B}_{MN} n^N = 0, \quad (\mathscr{E}_R)^M{}_{NP} = \mathscr{R}^M{}_{NPQ} n^Q, \quad (\mathscr{B}_R)^M{}_{NPQ} n^Q = 0. \tag{44}$$

The fact that $n$ generates a symmetry fixes the longitudinal parts $\mathscr{E}$ and $\mathscr{E}_R$ as

$$\mathscr{D}\mathscr{A}_- = \mathscr{E}, \qquad \mathscr{D}\left(\mathscr{D}_N n^M\right) = (\mathscr{E}_R)^M{}_N. \tag{45}$$

Then the hatted curvatures are

$$\hat{\mathscr{F}} = \mathscr{B} - \mathscr{A}_- F, \qquad \hat{\mathscr{R}}^M{}_N = (\mathscr{B}_R)^M{}_N - \mathscr{D}_N n^M F. \tag{46}$$

As a result, $u \wedge \hat{\mathscr{F}}$ and $u \wedge \hat{\mathscr{R}}$ differ from $u \wedge \mathscr{F}$ and $u \wedge \mathscr{R}$ by a term proportional to $F$.

Consequently, the formal $d + 4$ form

$$u \wedge \left(\mathscr{P} - \hat{\mathscr{P}}\right),$$

can be expanded as a formal power series in $F$ where each term has at least one $F$,

$$u \wedge \left(\mathscr{P} - \hat{\mathscr{P}}\right) = \sum_{i=0} V_i \wedge F^{i+1}, \tag{47}$$

and the $V_i$ are $d + 2(1 - i)$ forms. Similarly, the formal $d + 3$ form $u \wedge (I - \hat{I})$ can be written as a power series in $F$,

$$u \wedge \left(I - \hat{I}\right) = \sum_{i=0} W_i \wedge F^{i+1}, \tag{48}$$

where the $W_i$ are $d + 1 - 2i$ forms. Then we define the formal objects

$$\frac{u}{F} \wedge \left(\mathscr{P} - \hat{\mathscr{P}}\right) \equiv \sum_{i=0} V_i \wedge F^i, \qquad \frac{u}{F} \wedge \left(I - \hat{I}\right) \equiv \sum_{i=0} W_i \wedge F^i, \tag{49}$$

and in terms of these we find the compact expressions

$$\hat{V} = \frac{u}{F} \wedge \left(\mathscr{P} - \hat{\mathscr{P}}\right), \qquad \hat{W} = \frac{u}{F} \wedge \left(I - \hat{I}\right). \tag{50}$$

One can also obtain these from the direct integration of (42).

## 3.3 The anomalous boost Ward identity

We presently justify the assertions we made in (32) and (33) by explicitly computing the Milne variation of $\hat{V}$.

From (46) and (50) it is clear that we can regard $\hat{V}$ as $u$ wedge a functional of $\left\{\mathscr{B}, (\mathscr{B}_R)^M{}_N, F, \mathscr{A}_-, \mathscr{D}_N n^M\right\}$. Under an infinitesimal boost $u$ shifts as

$$\delta_\psi u = \psi, \qquad n^M \psi_M = 0. \tag{51}$$

Using (43) and that $\mathscr{F}$ and $\mathscr{R}$ are boost-invariant, this in turn induces

$$\delta_\psi \hat{\mathscr{A}} = -\mathscr{A}_-\psi\,, \qquad \delta_\psi(\hat{\Gamma}_g)^M{}_N = -(\mathscr{D}_N n^M)\psi\,, \qquad \delta_\psi F = d\psi\,, \tag{52}$$
$$\delta_\psi\hat{\mathscr{F}} = -\hat{\mathscr{D}}(\mathscr{A}_-\psi)\,, \qquad \delta_\psi\hat{\mathscr{R}}^M{}_N = -\hat{\mathscr{D}}\left(\mathscr{D}_N n^M \psi\right)\,,$$

where $\hat{\mathscr{D}}$ is the exterior covariant derivative defined using the hatted connections.

Although it is not obvious at this stage, these ingredients tell us that the computation of $\delta_\psi \hat{V}$ has already appeared in the literature. In the Appendix of [38], we and other authors computed the variations of a transgression form much like $\hat{V}$ where the difference between two connections was a one-form, just as is the case here. Rather than walk through that computation, let us simply quote the result,

$$\delta_\psi \hat{V} = d\left[\psi \wedge \frac{\partial \hat{V}}{\partial F}\right] - \delta_\psi\hat{\mathscr{A}} \wedge \cdot \frac{\partial \hat{\mathscr{P}}}{\partial \hat{\mathscr{F}}} - \delta_\psi(\hat{\Gamma}_g)^M{}_N \wedge \frac{\partial \hat{\mathscr{P}}}{\partial \hat{\mathscr{R}}^M{}_N}\,, \tag{53}$$

where the partial derivative of $\hat{V}$ is taken at fixed $\{u, \mathscr{A}_-, \mathscr{D}_M n^N, \mathscr{B}, \mathscr{B}_R\}$, and the objects

$$\frac{\partial \hat{\mathscr{P}}}{\partial \hat{\mathscr{F}}}\,, \qquad \frac{\partial \hat{\mathscr{P}}}{\partial \hat{\mathscr{R}}^M{}_N}\,, \tag{54}$$

are essentially the Hodge duals of the "Hall currents" that one gets by varying $\hat{I}$ with respect to the hatted connections $\hat{\mathscr{A}}$ and $\hat{\Gamma}_g$. However, both $\delta_\psi\hat{\mathscr{A}}$ and $\delta_\psi\hat{\Gamma}_g$ have no leg along $dx^-$, nor do the Hall currents in (54). As a result, the boost variation of the integral of $\hat{V}$, which is what actually appears in the anomaly inflow (31), only picks up the boundary term,

$$\delta_\psi \int_{\mathscr{M}_{d+2}} \hat{V} = \int_{\mathscr{M}_{d+1}} \psi \wedge \frac{\partial \hat{V}}{\partial F}\,. \tag{55}$$

This is what we meant when we made our earlier assertion that the boost variation of $\hat{V}$ is a boundary term (33). This also gives

$$G_\psi = \psi \wedge \frac{\partial \hat{V}}{\partial F}\,, \qquad \delta_\psi W' = -\int_{\mathscr{M}_{d+1}} G_\psi\,. \tag{56}$$

Using the epsilon tensor to dualize $\mathscr{Q} \equiv \frac{\partial \hat{V}}{\partial F}$ into a vector

$$q^M \equiv \frac{1}{d!}\varepsilon^{MP_1\dots P_d}\mathscr{Q}_{P_1\dots P_d}\,, \tag{57}$$

(here we have chosen an orientation such that $\varepsilon^{0\dots d-} = \frac{1}{\sqrt{\gamma}}$) the boost variation of the field theory generating functional becomes

$$\delta_\psi W' = -\int d^d x \sqrt{\gamma}\,\psi^\mu h_{\mu\nu}q^\nu\,, \tag{58}$$

where $\psi^\mu = h^{\mu\nu}\psi_\nu$. The anomalous boost Ward identity then reads

$$\mathscr{P}_\mu - h_{\mu\nu}J^\nu = h_{\mu\nu}q^\nu\,. \tag{59}$$

This has all been rather abstract. Let us see how this machinery works for pure $U(1)$ anomalies when the relativistic parent is two or four dimensional. In the first case, there is no Milne symmetry in the first place as there are no spatial directions, but we nevertheless proceed to illustrate how the machinery works. We have

$$\mathscr{P} = c_A \mathscr{F} \wedge \mathscr{F}\,, \tag{60}$$

and we readily find the formal objects

$$\hat{V} = c_A \mathcal{A}_- u \wedge (2\mathcal{B} - \mathcal{A}_- F), \qquad G_\psi = -c_A \mathcal{A}_-^2 \psi \wedge u. \tag{61}$$

But in this case $\psi = 0$ and so $G_\psi = 0$, so there is no Milne anomaly for the non-existent Milne symmetry.

In the four-dimensional case, we have

$$\mathscr{P} = c_A \mathscr{F} \wedge \mathscr{F} \wedge \mathscr{F}, \tag{62}$$

so that

$$\begin{aligned}
\hat{V} &= c_A \mathcal{A}_- u \wedge \left(3\mathcal{B} \wedge \mathcal{B} - 3\mathcal{A}_- F \wedge \mathcal{B} + \mathcal{A}_-^2 F \wedge F\right), \\
G_\psi &= -c_A \mathcal{A}_-^2 \psi \wedge u \wedge (3\mathcal{B} - 2\mathcal{A}_- F).
\end{aligned} \tag{63}$$

The anomalous boost Ward identity is

$$\mathscr{P}_\mu - h_{\mu\nu} J^\nu = -\frac{c_A \mathcal{A}_-}{2} h_{\mu\nu} \varepsilon^{\nu\rho\sigma} \left(3\mathscr{F}_{\rho\sigma} - 2\mathcal{A}_- F_{\rho\sigma}\right). \tag{64}$$

# 4 Weyl anomalies for $z = 2$

## 4.1 The basic idea

In what follows we will classify pure Weyl anomalies for Schrödinger theories. We do not presume or employ a Lagrangian description, but will simply use consistency conditions that all quantum field theories satisfy. We proceed in three steps.

1. First, we parameterize the most general local Weyl variation,

$$\delta_\Omega W = \int d^d x \sqrt{\gamma} \, \delta\Omega \, \mathscr{A},$$

   where $\mathscr{A}$ is a boost and gauge-invariant scalar built from the NC data $(n_\mu, h_{\mu\nu}, A_\mu)$ and derivatives.

2. Write down the most general set of local gauge/boost-invariant counterterms which may be added to $W$. Then compute the Weyl variation of these counterterms, and deduce which terms in $\mathscr{A}$ can be removed by a judicious choice of counterterm.

3. Impose Wess-Zumino (WZ) consistency [40], which in the present instance means that we demand

$$[\delta_{\Omega_1}, \delta_{\Omega_2}] = 0. \tag{65}$$

It should be clear that we can only perform this algorithm once we know the symmetries to consider and the background fields out of which we can build $\mathscr{A}$ and counterterms.

After performing this analysis, there are three classes of terms which appear in $\mathscr{A}$. We refer to terms which can be generated by local counterterms as class C, terms which are Weyl-covariant with weight $-(d+1)$ as class B, and terms which are not Weyl-covariant yet are WZ consistent as class A. (This is a different convention than that in [41].)

Because we consider pure Weyl anomalies, we need to efficiently classify boost and gauge-invariant scalars. As we mentioned in Subsection 2.2, the simplest way to do this is via the null reduction of a Lorentzian manifold with metric (12). In what follows we write down tensors in terms of the metric, isometry, and covariant derivative on this auxiliary higher-dimensional spacetime.

## 4.2 Two spatial dimensions

Let us warm up with the simplest non-trivial case, namely theories in $2+1$ dimensions. The $0+1$-dimensional case is trivial: in that case there are no tensors that may be formed from the NC structure, and so no potential Weyl anomaly.

For simplicity, we consider parity-preserving theories. Then the scalars that may contribute to $\mathscr{A}$ are built from the following basis of tensors: the inverse metric $g^{MN}$, the isometry $n^M$, the Riemann curvature $\mathscr{R}_{MNPQ}$, the derivative of $n$, $\mathscr{D}_M n_N$, and $\mathscr{D}_M$. Because $n^M$ generates a null isometry, we have

$$\mathscr{D}_{(M} n_{N)} = 0 \,, \qquad n^N \mathscr{D}_M n_N = 0 \,, \qquad \mathscr{R}_{MNPQ} n^Q = \mathscr{D}_P \mathscr{D}_N n_M \,. \tag{66}$$

Under constant Weyl transformations $e^\Omega = \lambda$, the basic building blocks transform as

$$g^{MN} \to \lambda^{-2} g^{MN} \,, \quad n^M \to n^M \,, \quad \mathscr{R}_{MNPQ} \to \lambda^2 \mathscr{R}_{MNPQ} \,, \quad \mathscr{D}_M n_N \to \lambda^2 \mathscr{D}_M n_N \,, \tag{67}$$

and $\mathscr{D}_M$ does not rescale. The putative Weyl anomaly transforms as $\mathscr{A} \to \lambda^{-4} \mathscr{A}$ under constant Weyl rescalings, and the local counterterms which can remove terms in $\mathscr{A}$ are the spacetime integrals of scalars $\mathscr{C}$ which also transform as $\mathscr{C} \to \lambda^{-4} \mathscr{C}$.

Scalars with weight $-4$ containing $p$ Riemann's, $q$ factors of the "twist" $\mathscr{D}_M n_N$, and $r$ additional covariant derivatives $\mathscr{D}_M$ necessarily contain $2+p+q$ factors of the inverse metric and $2(p-2)+r$ factors of $n^M$. In fact, we can only build such scalars if either $p \geq 2$ or $p = 1$ and $r \geq 2$. Such a scalar contains $2p + q + r$ derivatives. So the lowest number of derivatives possible is four, which can be done with either $p = 2, q = r = 0$ or $p = 1, q = 0$, and $r = 2$. In either case $n^M$ does not appear. That is, the scalars of weight $-4$ with the lowest number of derivatives are just the ordinary four-derivative scalars which can be built from the Riemann, the covariant derivative, and the metric. We parameterize them using the basis

$$E_4 = \mathscr{R}^{MNPQ} \mathscr{R}_{MNPQ} - 4 \mathscr{R}^{MN} \mathscr{R}_{MN} + \mathscr{R}^2 \,, \quad \mathscr{W}^{MNPQ} \mathscr{W}_{MNPQ} \,, \quad \mathscr{R}^2 \,, \quad \mathscr{D}_M \mathscr{D}^M \mathscr{R} \,, \tag{68}$$

where $\mathscr{R}_{MN} = \mathscr{R}^P{}_{MPN}$ is the Ricci curvature, $E_4$ is the four-dimensional Euler density built from $\mathscr{R}_{MNPQ}$, and $\mathscr{W}_{MNPQ}$ the Weyl tensors. We have also used that $\mathscr{D}_M \mathscr{D}_N \mathscr{R}^{MN} = \frac{1}{2} \mathscr{D}_M \mathscr{D}^M \mathscr{R}$ by the Bianchi identity.

Due to (66) there are no weight $-4$ scalars with five derivatives. There are a number of six-derivative scalars, like

$$\mathscr{R}^2 \mathscr{R}_{MN} n^M n^N \,.$$

At this point, it should be clear that the problem of classifying the four-derivative terms in the Weyl anomaly is just the same problem as for $3+1$-dimensional relativistic CFT. The Euler and $\mathscr{W}^2$ counterterms are Weyl-invariant up to a boundary term, the $\mathscr{R}^2$ counterterm can be used to remove the $\mathscr{D}_M \mathscr{D}^M \mathscr{R}$ term in $\mathscr{A}$, and the $\mathscr{D}_M \mathscr{D}^M \mathscr{R}$ counterterm integrates to a boundary term. The remaining $E_4$ and $\mathscr{W}^2$ terms in $\mathscr{A}$ are WZ consistent, but the $\mathscr{R}^2$ term is not. So we have

$$\mathscr{A} = a E_4 - c \mathscr{W}^2 + \mathscr{O}(\partial^6) \,. \tag{69}$$

The four-derivative part is just the usual Weyl anomaly of a $3+1$-dimensional CFT on the background (12). The Euler term is a class A anomaly, the $\mathscr{W}^2$ term is class B, and the $\mathscr{D}_M \mathscr{D}^M \mathscr{R}$ term is class C.

It is worth noting that even though there is an isometry both $E_4$ and $\mathscr{W}_{MNPQ}$ can be nonzero. However, if the isometry has no twist, $\mathscr{D}_M n_N = 0$, then (66) implies that $\mathscr{R}_{MNPQ}$ is effectively the Riemann tensor of a three-dimensional space, in which case $E_4$ and $\mathscr{W}_{MNPQ}$ vanish. In the coordinates (12), the nonzero components of the twist are

$$\mathscr{D}_\mu n_\nu = \frac{1}{2} (\partial_\mu n_\nu - \partial_\nu n_\mu) \,. \tag{70}$$

So the Weyl anomaly is only visible when $dn \neq 0$.

What of the potential higher-derivative terms in $\mathscr{A}$? While we do not rigorously classify them, we offer some observations which lead to a conjecture for $\mathscr{A}$.

First, by WZ consistency, any scalar which transforms covariantly with weight $-4$ under inhomogeneous Weyl transformations may appear in $\mathscr{A}$. If $n$ was everywhere timelike or spacelike rather than null, then we could use it to redefine the covariant derivative in a Weyl-invariant way. Then the Weyl-covariant tensors would be built from this Weyl-covariant derivative. However, because $n$ is null no such redefinition of $\mathscr{D}_M$ exists. So the manifestly Weyl-covariant tensors are built from $\mathscr{W}_{MNPQ}$ along with $g^{MN}$ and $n^N$. We also expect there to be non-manifest Weyl-covariant tensors which cannot be expressed this way. It seems that one can build weight $-4$ scalars with arbitrarily many derivatives from this data by a similar counting argument to that above. The manifestly Weyl-covariant scalars can be counted as follows. A $2p$-derivative scalar possesses $p$ Weyl tensors along with $2+p$ factors of the inverse metric and $2(p-2)$ factors of $n^M$ (with $p \geq 2$). For example, a six-derivative scalar is

$$\mathscr{W}_{MNPQ}\mathscr{W}^{MNP}{}_S\mathscr{W}^Q{}_A{}^S{}_B n^A n^B \, . \tag{71}$$

We have not found an argument that there are a finite number of such scalars.

Second, the Weyl variation of any local weight $-4$ counterterm is always of the form (we can represent the counterterm equivalently as a 4-dimensional integral or as a 3-dimensional one, since no sources depend on the null circle)

$$\delta_\Omega \int d^3x \sqrt{\gamma}\,\mathscr{C} = \int d^3x \sqrt{\gamma}\,\delta\Omega\,\mathscr{D}_M\mathscr{C}^M + (\text{boundary terms})\, . \tag{72}$$

With this in mind, experience has taught us that counterterms can be used to remove all terms in $\mathscr{A}$ of the form $\mathscr{D}_M\mathscr{A}^M$, although we lack a proof that this is always the case.

Third, we conjecture that $E_4$ is the unique "exceptional" scalar built from $g_{MN}$ and $n^M$ and derivatives which is not Weyl-covariant, yet is WZ consistent when it appears in $\mathscr{A}$.

Putting all three of these ingredients together, we can make a conjecture for $\mathscr{A}$, namely that it is of the form

$$\mathscr{A} = aE_4 - c\mathscr{W}^2 + \sum_i d_i\mathscr{W}_i^n \, , \tag{73}$$

where the $\mathscr{W}_i^n$ are Weyl-covariant scalars with at least six derivatives, built using $n^M$.

### 4.3 The general result

The general argument proceeds in the same way as that above.

First, consider a Schrödinger theory in even spacetime dimension $d$. We build an auxiliary odd-dimensional metric and isometry from the NC structure. The Weyl anomaly $\mathscr{A}$ must transform with weight $d+1$ under constant Weyl transformations, which in this instance means it transforms with odd weight. However, there is simply no way to assemble an odd weight scalar out of the building blocks (67). So $\mathscr{A} = 0$ in this case.

In odd spacetime dimension $d = 2m-1$, we can build weight $-(d+1)$ scalars with $p$ Riemanns, $q$ factors of the "twist" $\mathscr{D}_M n_N$, and $r$ derivatives, provided that we contract indices with $2+p+q$ factors of the inverse metric and $2(p-m)+r$ factors of $n^M$. Such a scalar possess $2p+q+r$ derivatives. The terms with the lowest number of derivatives have $2p+r = m$ and $q = 0$, in which case no factors of $n^M$ appear. So, as in the $2+1$-dimensional case, the lowest-derivative analysis reduces to the relativistic one, for which we can appeal to the literature [41],

$$\mathscr{A} = aE_{d+1} + \sum_n c_n\mathscr{W}_n + \mathcal{O}(\partial^{d+3})\, , \tag{74}$$

where the $\mathscr{W}_n$ are weight $-2m$ Weyl-covariant scalars, and $E_{d+1}$ is the $d+1$ dimensional Euler density.

We conjecture that all of the higher-derivative terms are Weyl-covariant scalars $\mathscr{W}_i^n$ which depend on $n^M$, giving

$$\mathscr{A} = aE_{d+1} + \sum_n c_n \mathscr{W}_n + \sum_i d_i \mathscr{W}_i^n. \tag{75}$$

A BRST-inspired analysis [42] should be able to verify or disprove our conjecture.

# Acknowledgments

We are pleased to thank J. Hartong for useful discussion. The author would like to thank the organizers of the "2014 Simons Summer Workshop in Mathematics and Physics" at the Simons Center for Geometry and Physics for their hospitality during which most of this work was completed. The author was supported in part by National Science Foundation under grant PHY-0969739.

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
