# Peer review of "Anomalies for Galilean fields"

_SciPost Physics, doi:SciPost Phys. 5, 005 (2018)_

## Round 1 · Referee Report · Anonymous · 2018-3-11

Strengths

1. well-written
2. interesting and timely subject
3. useful results for future research

Weaknesses

1. paper came out as arXiv several years ago, and there have been further developments in this research topic

Report

This is an interesting and well-written paper on anomalies in field theories with Galilean symmetry, including those
that also have an anisotropic z=2 scaling symmetry (i.e. Schroedinger symmetry).
Two aspects are studied. The first one is to consider the null reduction (discrete light-cone quantization) of relativistic field theories
that have flavor or gravitational anomalies and then examining whether these anomalies survive the null reduction.
It is shown that these become mixed flavor/boost or gravitational/boost anomalies.
In a second direction the pure Weyl anomalies of z=2 Schroedinger theories are considered showing that are no such anomalies in even spacetime dimensions, while in odd dimensions (restricting to the lowest derivative ones) these are in 1-1 correspondence to those of a
relativistic CFT in one dimension higher.

There has been in the last five years a renewed interesting in aspects of non-relativistic physics, coming from a variety of directions ranging from applications to condensed matter systems, non-AdS holography and non-relativistic gravity theories and string theory. This paper
addresses an important aspect of non-relativistic field theories, namely the question of anomalies. Given the quality of the paper and the novelty of the results I can recommend it to be published though the author should take care of the following two points:

Requested changes

1. This paper came out as an arXiv preprint in 2014, i.e. already quite some years ago, and since that time a number of further important papers have appeared on this topic (which in fact refer to the present paper). If this paper will be published (as I recommend) in Sci-Post in 2018, it would be odd to ignore in the published version the main developments that have occurred since then. Therefore I suggest the author
to include (e.g. in a note added, or as an extra paragraph in the intro our conclusion) one paragraph summarizing these main developments
at least those that are directly related to the results of the present paper. In particular, how the results fit with those of 1601.06795 (JHEP 1606 (2016) 158) and also the earlier paper 1511.08150 (JHEP 1602 (2016) 003, Erratum: JHEP 1602 (2016) 177) by another group of authors, which in the first version overlooked an aspect, as later pointed out in 1601.06795. There might be further papers on non-relativistic anomalies that the author wishes to include in this part.

2. The author refers to Ref. [5] for the covariant coupling of non-relativistic field theories to Newton-Cartan geometry
(e.g. below eq. (2.4)). Here a reference to the article 1409.1519 (Phys.Lett. B746 (2015) 318-324) which obtained this independently
at the same time, should be included as well.

---

## Round 1 · Referee Report · Anonymous · 2018-3-19

Strengths

1. The paper is well written, concise and clear about the assumptions and results.

2. The results are useful for further developing our understanding of NR field theories and their holographic descriptions.

Weaknesses

1. Whereas on the Newton-Cartan side the paper is rather self-contained, this is not the case for the study of anomalies. Results from previous papers are copied without much explanation.

2. The results are limited to anomalies that can be understood from null reduction.

Report

This manuscript appeared on the arxiv a bit more than 3 years ago. The analysis is a straightforward application of the technique of null reduction to anomalies in relativistic theories. Ultimately one would like to have an intrinsic understanding of the anomaly structure of NR field theories without adhering to a relativistic parent theory (provided it exists). I consider this preprint an important first step in the direction of understanding the possible anomalies in NR field theories and I therefore recommend it for publication.

Requested changes

1. Three years have passed since this preprint was written, so the author should place the results in the context of what we know today about NR anomalies.

2. There appears to be a small typo in section 3.1. Just above eq. (3.1) a lower case d should be capital D. From section 3.2 we have D=d+1.

  • validity: high
  • significance: high
  • originality: high
  • clarity: high
  • formatting: perfect
  • grammar: perfect

Author:  Kristan Jensen  on 2018-06-08  [id 270]

(in reply to Report 2 on 2018-03-19)

I would like to begin by thanking both referees for their comments. I of course agree that the publication situation is a bit awkward. Per the referees' suggestion, I have added a paragraph at the end of the Introduction which summarizes what I consider to be the two most interesting developments viz. Galilean anomalies since this paper originally appeared on the arXiv. Namely, the independent classification of Galilean anomalies, and the direct computation of the anomaly of a free NR scalar field.

There are actually three different computations of that anomaly by three different groups, all of which find different results. For reasons I can elaborate on, I believe the computation of Grinstein and Pal, and gave one reason why in the "Note added."

Finally, regarding referencing, I added the desired reference to Hartong et al in a Footnote in the Introduction, although I would point out that their construction differs a bit from the one used in my note.

Attachment:

anomalies_v2.pdf

---

## Round 2 · List of Changes

Per the referees' suggestion, I added a brief note at the end of the Introduction summarizing some of the main developments viz. anomalies in Galilean field theories since my note originally appeared on arXiv.

You are currently on this page

Resubmission 1412.7750v2 on 18 June 2018

---

## Editorial Decision

published